# Balanced Token Pruning: Accelerating Vision Language Models Beyond Local Optimization

**Kaiyuan Li**[1*], **Xiaoyue Chen**[1*], **Chen Gao**[2†], **Yong Li**[2], **Xinlei Chen**[1†]

[1]Tsinghua Shenzhen International Graduate School
[2]BNRist, Tsinghua University

{likaiyua23,chenxiao24}@mails.tsinghua.edu.cn
{chgao96,liyong07}@tsinghua.edu.cn,chen.xinlei@sz.tsinghua.edu.cn

## Abstract

Large Vision-Language Models (LVLMs) have shown impressive performance across multi-modal tasks by encoding images into thousands of tokens. However, the large number of image tokens results in significant computational overhead, and the use of dynamic high-resolution inputs further increases this burden. Previous approaches have attempted to reduce the number of image tokens through token pruning, typically by selecting tokens based on attention scores or image token diversity. Through empirical studies, we observe that existing methods often overlook the joint impact of pruning on both the current layer's output (local) and the outputs of subsequent layers (global), leading to suboptimal pruning decisions. To address this challenge, we propose Balanced Token Pruning (BTP), a plug-and-play method for pruning vision tokens. Specifically, our method utilizes a small calibration set to divide the pruning process into multiple stages. In the early stages, our method emphasizes the impact of pruning on subsequent layers, whereas in the deeper stages, the focus shifts toward preserving the consistency of local outputs. Extensive experiments across various LVLMs demonstrate the broad effectiveness of our approach on multiple benchmarks. Our method achieves a 78% compression rate while preserving 96.7% of the original models' performance on average. Our code is available at https://github.com/EmbodiedCity/NeurIPS2025-Balanced-Token-Pruning.

## 1 Introduction

Recent advances in Large Vision-Language Models (LVLMs) [8, 15, 27, 29, 43, 52] have substantially improved visual understanding. These models typically employ a visual encoder to convert images into discrete tokens, which are then processed jointly with textual tokens by a large language model backbone. The incorporation of visual information significantly increases the total number of input tokens [2, 28, 59], a problem further amplified when handling high-resolution images. In edge applications such as emergency monitoring [7, 44], logistics [6, 56], and smart homes [51], models are typically deployed on devices like drones and unmanned vehicles [23, 10], which are constrained by limited memory and strict latency requirements. The excessive number of image tokens poses a major bottleneck for deployment, drawing increasing research interest in accelerating edge inference [38].

Prior studies [1] have demonstrated that visual tokens often exhibit significant redundancy [5, 26]. Consequently, visual token pruning has been proposed as an effective strategy to reduce input redundancy and enhance computational efficiency [47, 40, 18, 53, 48]. Visual token pruning faces

---

*Equal Contribution.
†Corresponding author.

39th Conference on Neural Information Processing Systems (NeurIPS 2025).

two fundamental challenges: identifying the most important visual tokens and determining the appropriate layers for pruning. Existing token pruning strategies can be broadly classified into two categories: attention-based methods that leverage text-image interactions [5, 46, 32], and diversity-based methods that exploit the heterogeneity of visual representations [1]. However, the impact of their distinct optimization objectives on overall model performance remains underexplored, and a systematic comparison between them is largely absent. Moreover, when it comes to pruning layer selection, existing methods rely heavily on validation performance and manually defined settings, lacking principled guidance based on the model's intrinsic properties.

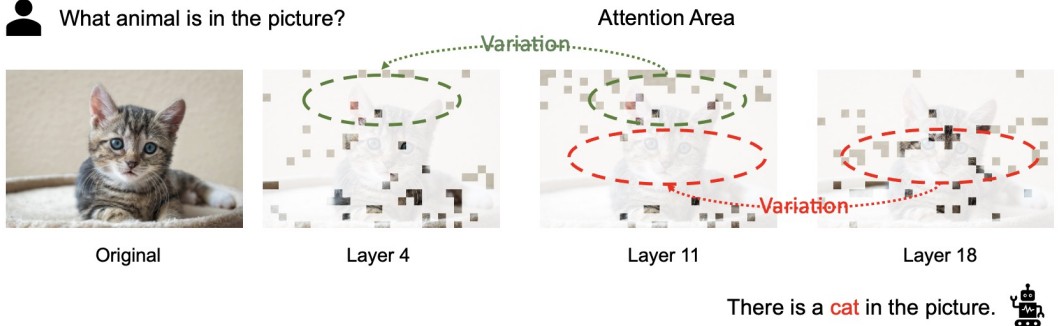

Figure 1: Layer-wise visualization of attention in LVLMs.

To address these problems, we first explore the nature of image token pruning from an intuitive perspective: its impact on the **current layer's (local)** output and its influence on the outputs of **subsequent pruning layers (global)**. We begin by visualizing the spatial distribution of image tokens that receive higher attention from text tokens across different layers. As shown in Figure 1, we observe that the image tokens attended by text tokens vary across different layers. This indicates that pruning solely based on the current layer tends to overlook its impact on subsequent layers. Then we further investigate the impact of different pruning methods on the model outputs. Specifically, we compare the hidden states of output tokens at different decoding positions under two pruning methods with those of the original model.

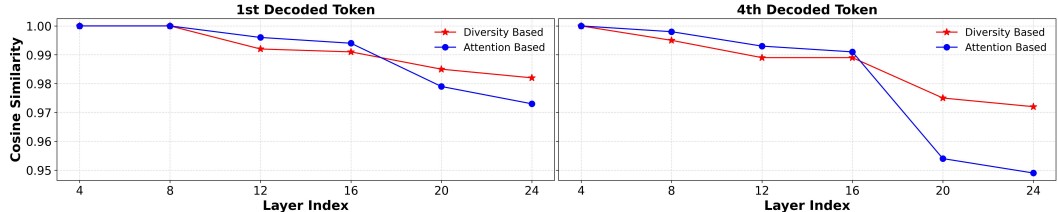

Figure 2: Impact of different pruning strategies on layer-wise representations.

It can be found in Figure 2 that attention-based methods preserve output similarity well at early pruning layers, but the error accumulates in deeper layers. In contrast, diversity-based methods do not maintain output similarity at the initial layers, but achieve better consistency in later pruning stages. This implies that attention-based pruning methods focus solely on optimizing the current pruning layer while ignoring their impact on subsequent layers, whereas diversity-based methods overlook the preservation of output quality at the current layer.

Motivated by the above observation, we aim to tackle a fundamental challenge: *how to prune with joint consideration of the current and subsequent layers to achieve global optimality.* To address this challenge, we propose **Balanced Token Pruning (BTP)**, a visual token pruning method that balances local objectives (current layer) with global objectives (subsequent layers). We begin by analyzing and formulating a local-global objective for image token pruning. Based on this objective, BTP first partitions the pruning process into multiple stages using a small calibration set [35, 20], leveraging the way LVLMs process images, as illustrated in Figure 4. In early stages, where more image tokens are retained, BTP emphasizes a diversity-based objective to preserve the quality of downstream representations. In later stages, where fewer tokens are retained, it prioritizes an attention-based objective to maintain the consistency of local outputs. With this design, we preserve token diversity in the early layers while focusing on task-relevant tokens in the later layers.

Extensive experiments demonstrate the effectiveness of our proposed BTP method. We evaluate BTP across models of varying sizes and architectures, consistently achieving superior performance under higher compression ratios. Notably, our approach retains only 22% of the original image tokens on average while preserving 98% of the model's original performance. Furthermore, end-to-end efficiency evaluations confirm that BTP significantly reduces both inference latency and memory usage in practice.

## 2 Related work

### 2.1 Large Vision-Language Models

Recent progress in large vision language models (LVLMs) has been substantially accelerated by the open-sourcing of foundation models like LLaMA [41] and Vicuna [60]. Representative models, including LLaVA [27, 28, 29], Qwen-VL [2, 43], and InternVL [8, 15] leverage vision encoders [34, 25, 9] to encode images into visual tokens, which are then integrated into the language model for unified multimodal representation and understanding [14]. For example, LLaVA-1.5 encodes image into 576 visual tokens using a single-scale encoder. As these models increasingly support high-resolution visual inputs [2, 28], the number of visual tokens grows. Using a multi-resolution encoding strategy, LLaVA-NeXT can generate up to 2,880 tokens per image. Multimodal large models have been widely applied in various scenarios, including embodied agent [17]. The large number of image tokens limits its applicability in scenarios such as real-time applications [39].

### 2.2 Visual Token Pruning

Early efforts to reduce visual token redundancy primarily focus on attention-based pruning [4, 19, 54, 32]. For example, FastV [5] prunes visual tokens with low attention scores after the filtering layer, with subsequent layers processing only the remaining token. Another approach, VTW [26], adopts a complete token elimination strategy, removing all visual tokens after a specified layer. PyramidDrop [46] introduces a more sophisticated approach, performing stage-wise pruning throughout the transformer, ranking visual tokens by their attention scores to the instruction token at each stage and progressively discarding the least informative ones. Compared to attention-based methods, diversity-based methods prioritize retaining a richer variety of semantic information. For instance, DivPrune [1] formulate token pruning as a Max-Min Diversity Problem [33, 37]. Additionally, some methods fuse remaining tokens into retained tokens through token fusion such as LLaVA-PruMerge [40] and VisionZip [47]. Different from prior methods, our method jointly considers the impact of pruning on both the current layer and subsequent layers.

## 3 Preliminary

### 3.1 Visual token processing

In the prefilling stage, images and texts are first encoded into embedding vectors (tokens), which are then processed by LVLM. We denote the input token sequence as $\mathbf{X}$ which consists of the system prompt $\mathbf{X}_S$, the image tokens $\mathbf{X}_I$ and text tokens $\mathbf{X}_T$, $\mathbf{X} = (\mathbf{X}_S, \mathbf{X}_I, \mathbf{X}_T)$. $\mathbf{X}$ is then fed into the LLM backbone composed of $N$ decoder layers. For the $l$-th decoder layer, we denote the input as $\mathbf{X}^{(l)}$ and the layer output $\mathbf{X}^{(l+1)}$ is:

$$\mathbf{X}^{(l+1)} = \mathbf{X}^{(l)} + Atten^{(l)}(LN(\mathbf{X}^{(l)})) + \text{MLP}^{(l)}(LN(attn_{output}^{(l)} + \mathbf{X}^{(l)})), \tag{1}$$

where $Atten^{(l)}$ is the attention block, $LN$ is the layer normalization and $MLP^{(l)}$ is the projector layer. It can be observed that the outputs of the attention block and the MLP block are closely tied to the attention mechanism [42]. Formally, the attention mechanism can be represent as:

$$attn_{output}^{l} = Softmax(\frac{Q_l(K_l)^T + M}{\sqrt{D_k}})V_l, \tag{2}$$

where $Q_l$, $K_l$, $V_l$ are calculated by Query projector, Key projector and Value projector. $D_k$ is hidden state dimension. $M$ is the casual mask which imposes a constraint such that each token is permitted to incorporate information only from tokens at earlier positions. $K_l$, $V_l$ are stored in the KV cache for further decoding stage.

## 3.2 Visual token pruning formulations

Prior works on image token pruning can be broadly categorized into attention-based methods [5, 46] and diversity-based methods [1]. Attention based methods utilize text-image attention score to select important image tokens at specific layers. For input sample with $m$ text tokens, we can denote the importance score $S_{img}$ of image tokens at $l$-th layer as:

$$S_{img}^{(l)} = \frac{1}{m} \sum_{i=1}^{m} Atten^{(l)}(\mathbf{X}_I, \mathbf{X}_T^{(i)}).$$ (3)

After obtaining the importance scores of the image tokens, these methods select a pruned image token set $\mathbb{P}_{atten} \subset \mathbf{X}_I$ with the highest scores. In contrast to attention score-based methods, diversity-based approaches focus on maximizing the diversity among selected image tokens. These methods are typically based either on the spatial diversity of the selected image token set or on the semantic diversity of the selected images. Formally, given a diversity metric $\mathcal{F} \subset \{\mathcal{F}_{spa}, \mathcal{F}_{sem}\}$, our goal is to identify a pruned set of image tokens $\mathbb{P}_{div} \subset \mathbf{X}_I$ that maximizes the objective function $\mathcal{L}_{div}$:

$$\mathcal{L}_{div} = \max \mathcal{F}(\mathbb{P}_{div}).$$ (4)

# 4 Methodology

## 4.1 Limitations of existing methods

**Attention-based methods pursue local optima**   We analyze the impact of pruning image tokens on the subsequent text and response tokens. From Equations 1 and 2, we can see that pruning image tokens at $l$-th layer mainly affects the layer output $\mathbf{X}^{(l+1)}$ by changing the attention output, which is a weighted sum of the value vectors $V_l$. If the norms of the $V_l$ are similar, selecting image tokens with high importance scores defined in 3 effectively reduces the difference between the layer output before and after pruning. We provide supporting evidence for this assumption in the Appendix 7.1. Formally, given original $l$-th layer output $\mathbf{X}_{origin}^{(l+1)}$ and pruned $l$-th layer output $\mathbf{X}_{pruned}^{(l+1)}$, distance metric function $D(\cdot, \cdot)$, we can define the objective function $\mathcal{L}_{atten}$ of attention-based methods [5, 46] as:

$$\mathcal{L}_{atten} = \min_{\mathbb{P}} D(\mathbf{X}_{origin}^{(l+1)}, \mathbf{X}_{pruned}^{(l+1)}).$$ (5)

However, attention-based methods locally optimize the output error at individual layers. For instance, if pruning is conducted at the $l$-th layer and $(l+k)$-th layers, with $\mathbb{P}_l$ and $\mathbb{P}_{l+k}$ denoting the respective optimal sets of selected image tokens. As shown in Figure 1, $\mathbb{P}_{l+k} \not\subset \mathbb{P}_l$. So, attention-based selection results in a **globally suboptimal** pruning strategy.

**Diversity-based methods ignore local constraints**   The diversity-based approach [1] aims to maximize the diversity of the selected tokens, thereby partially mitigating the issues encountered by attention-based methods as we can see in Figure 1. Because diversity-based methods tend to select tokens with maximally different semantic information. However it can be observed in Figure 2 that diversity-based approaches are ineffective in maintaining local output consistency, which can lead to a failure in preserving **local output consistency** during deep-layer pruning, resulting in degraded performance.

**Layer selection for pruning**   Current approaches typically rely on manually predefined pruning layers or utilize a small validation set to select pruning layers based on the observed performance. However, these methods require extensive trial-and-error and dataset-specific calibration. As described in Section 3.1, due to the presence of the causal mask $M$, the encoding of an image token in the LLM backbone is independent of the input question. Therefore, we aim to determine the pruning layers from the perspective of image token encoding.

## 4.2 Balanced token pruning with joint local and global objectives

**Local-global objective**   Based on the above analysis, we argue that an effective token pruning strategy should achieve local optimization by preserving the current layer's output, while also considering the global impact of pruning on subsequent layers. As shown in Equation 1, the model's output depends on both the outputs of previous layers and the attention module of the current

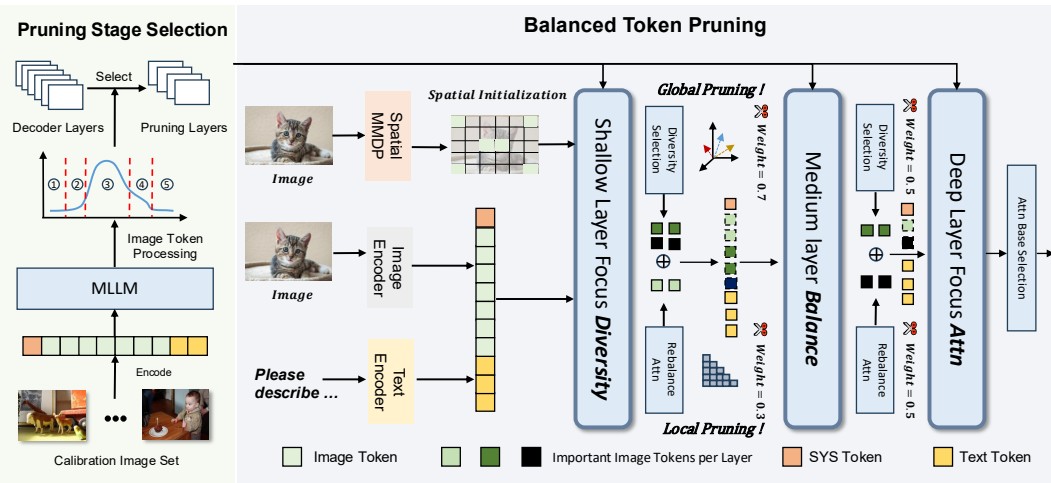

Figure 3: Overview of BTP: We first use a calibration set to determine the pruning layers. In the early layers, we emphasize diversity-based pruning to preserve the output of subsequent layers. In the deeper layers, attention-based pruning is prioritized to maintain the output of the pruning layers. Due to the pruning strategy, we achieve an overall optimal pruning balance.

layer. Therefore, to ensure that the final output of the pruned model remains similar to that of the original model, we should maintain the similarity between the output of each pruned layer and its corresponding original output. Firstly, we formulate a **global objective** function. Suppose token pruning is performed at layers $l_1 < l_2 < l_3$. For each pruned layer $l \in \{l_1, l_2, l_3\}$, we aim to select a subset of tokens $\mathbb{P}_l$ such that the difference between the pruned outputs $X_{pruned}^{l+1}$ and original outputs $X_{origin}^{l+1}$ is minimized. To quantify hidden vectors' difference, we use a unified distance function $D(\cdot, \cdot)$ to measure the discrepancy between the outputs before and after pruning. Then our objective is to minimize the total output discrepancy across all pruned layers:

$$\mathcal{L}_{global} = \sum_{i=1}^{|l|} D(X_{origin}^{l+1}, X_{\mathbb{P}_{l_i}}^{l+1}). \tag{6}$$

According to Equation 5, we can get optimal pruned token set $\mathbb{P}_l^*$ based on attention. However, since the attention distribution varies across input samples and $P_{l_3} \subseteq P_{l_2} \subseteq P_{l_1}$, it is difficult to predict which tokens will be important for deeper layers (e.g., $l_2, l_3$) when pruning at layer $l_1$. To address this issue, we propose to optimize a **local-global objective** to approximate the optimal token set $P_l^*$. Building upon the local attention-based selection objective, we introduce a diversity term to approximate the token preferences of later layers. Assume a weight coefficient $\lambda \in (0, 1)$, we measure diversity by computing the sum of distance $F_{dis}(\cdot)$ among elements within a set:

$$\mathcal{L}_{local-global} = -\sum_{i=1}^{|l|} (\lambda_i \sum_{j \in P_i} Atten^{(i)}(\mathbf{X}_I^{(j)}, \mathbf{X}_T) + (1 - \lambda_i) F_{dis}(P_i)). \tag{7}$$

The first term of Equation 7 ensures that the output of the pruned layer remains close to the original, while the second term encourages the selected tokens at previous layer $l_1$ to also include those important for deeper layers such as $l_2$ and $l_3$.

**Balanced token pruning (BTP)** Building upon the proposed local-global objective, we introduce our method. Figure 3, our approach divides token pruning into multiple stages denoted as $\mathcal{S} = \{s_1, \ldots, s_n\}$. Under a predefined pruning ratio $\alpha$, each stage retains a fixed fraction of image tokens from the previous stage. As shown in Appendix 7.2, we can observe that retaining only a small number of image tokens is sufficient to optimize the attention objectives. Since early pruning stages retain more tokens and influence the pruning decisions of later stages, their objectives need to emphasize token diversity. In contrast, deeper stages preserve fewer tokens and have less impact on subsequent stages. Therefore, we set the hyperparameter $\lambda_i$ to gradually increase across stages.

**Attention optimization:** We optimize the attention objective by selecting the top-$k$ image tokens with the highest importance scores defined in Equation 3. To efficiently computing the importance scores, we only use the last token of the input prompt as $\mathbf{X}_T$, which reduces the computational complexity to $\mathcal{O}(n)$. We observe that the attention scores are influenced by positional encoding, which leads to a tendency to favor tokens located toward the end of the sequence. We apply a re-balancing operation to alleviate the influence of positional encoding. Assume that at $l$-th layer, we aim to prune the image tokens by selecting $k$ indices $I_k$ out of $N$ candidates based on the attention scores $A_l$. Instead of directly selecting the top-$k$ tokens, we first over-select the top-$k'$ tokens indices $I_{k'}$, where $k' > k$. To mitigate positional bias, we rebalance the selection by first retaining tokens from earlier positions, followed by selecting additional tokens from later positions:

$$I_{pre} = I_{k'}[I_{k'} < \frac{N}{2}], \tag{8}$$

$$I_{post} = I_{k'}[I_{k'} \geq \frac{N}{2}][: k - |I_{pre}|], \tag{9}$$

$$I_k = Concat(I_{pre}, I_{post}). \tag{10}$$

Through the rebalancing operation, we are able to preserve the attention objective while selecting more informative tokens.

**Diversity optimization:** For optimizing the second objective related to diversity, we follow the formulation used in DivPrune by modeling it as a Max-Min Diversity Problem (MMDP). However, solving the MMDP objective requires $\mathcal{O}(n^2)$ computational complexity and cannot be efficiently accelerated by GPUs, resulting in significant computational latency. This issue becomes more pronounced in high-resolution multimodal models with a larger number of image tokens. To address this challenge, we propose an initialization strategy based on spatial position information. We observe that image patches with large spatial distances tend to exhibit greater semantic differences, while spatially adjacent patches are often semantically similar. Based on this intuition, we initialize the set of selected image tokens by solving an MMDP problem over their spatial positions. Formally, given $N$ image tokens $\mathbf{X}_I$, which are originally obtained by flattening a 2D image, we first formulate a 2D grid of size $\sqrt{N} \times \sqrt{N}$. For any two tokens $y$ and $w$ from the $N$ tokens, their distance is defined as the Manhattan distance $d(\cdot, \cdot)$ between their positions in the 2D grid. Based on this distance metric, we construct the initial token set $E_{initial}$:

$$E_{initial} = argmax[\min_{y,w \in S}(d(y, w) : \forall S \subset \mathbf{X}_I]. \tag{11}$$

### 4.3 Pruning layer selection

We propose that determining which layers to prune is closely related to encoding process of image tokens. Specifically, pruning should occur either before or after the layers where the meaning of image tokens changes significantly, since it is difficult to identify truly important tokens in such layers. We compute the cosine similarity between image token hidden states $X_I^l, X_I^{l+1}$ before and after each layer. For each layer, we plot the number of tokens with similarity below threshold $\tau$ alongside the total attention allocated to image tokens. As shown in Figure 4, it can be observed that LVLMs tends to allocate more attention to image tokens in layers following those where the representations of image tokens undergo significant changes. Based on these insights, we propose a task-independent layer selection strategy for pruning. Using a fixed set of 64 samples across all datasets, we identify layers immediately before and after major shifts in image token semantics. As shown in Figure 3, we perform pruning at selection layers, which enhances the effectiveness of our pruning strategy.

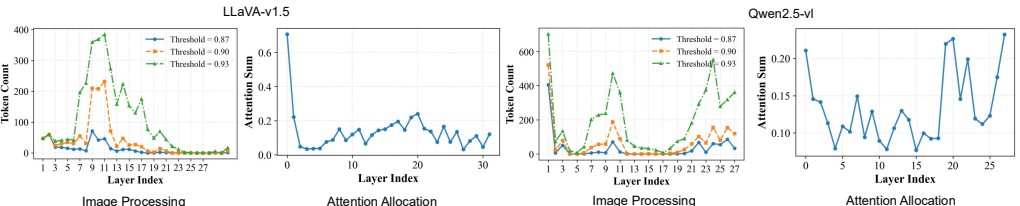

Figure 4: Layer-wise image token hidden state dynamics and attention allocation in LVLMs.

Table 1: Comparison of BTW with VTW, PDrop, FastV, and DivPrune across different models and datasets. * : For models using dynamic resolution, we report the token retention ratio instead of the absolute token count.

| Method | Token | TFLOPS | GQA | MME | MMB$_{en}$ | POPE | SQA | MMVET | Avg. |
|---|---|---|---|---|---|---|---|---|---|
| **LLaVA-1.5-7B** | | | | | | | | | |
| Original | 576 | 3.82 | 62.0 | 1510.7 | 64.3 | 85.8 | 69.4 | 29.0 | 100% |
| VTW (AAAI25) [26] | 236 | 1.67 | 51.3 | 1475.0 | **63.4** | 82.1 | 68.8 | 17.8 | 89% |
| PDrop (CVPR25) [46] | 192 | 1.30 | 57.1 | 1399.0 | 61.6 | 83.6 | 68.4 | 25.8 | 94% |
| FastV (ECCV24) [5] | 172 | 1.65 | 57.6 | 1465.0 | 61.6 | 81.0 | 68.9 | **29.3** | 96% |
| DivPrune (CVPR25) [1] | 128 | 0.83 | 58.8 | 1405.4 | 62.1 | 85.1 | 68.4 | 27.4 | 96% |
| **BTP**(ours) | 128 | 0.85 | **59.0** | 1487.0 | 62.7 | **85.6** | 69.1 | 29.1 | **98%** |
| **LLaVA-1.5-13B** | | | | | | | | | |
| Original | 576 | 7.44 | 63.2 | 1521.7 | 68.8 | 87.0 | 72.7 | 37.4 | 100% |
| VTW (AAAI25) [26] | 236 | 2.97 | 55.6 | 1517.1 | 67.7 | 79.0 | 72.2 | 22.6 | 89% |
| PDrop (CVPR25) [46] | 192 | 2.46 | 60.5 | 1493.0 | 67.3 | 85.1 | **73.7** | 32.8 | 96% |
| FastV (ECCV24) [5] | 172 | 2.25 | 60.0 | 1473.0 | 67.0 | 83.6 | 72.9 | 31.9 | 95% |
| DivPrune (CVPR25) [1] | 128 | 1.63 | 58.8 | 1461.0 | 65.8 | 86.5 | 72.6 | 34.0 | 96% |
| **BTP**(ours) | 128 | 1.68 | **62.2** | **1519.7** | **68.0** | **86.9** | 72.7 | **34.5** | **98%** |
| **LLaVA-1.6-7B *** | | | | | | | | | |
| Original | 100% | 20.82 | 64.2 | 1519.3 | 67.1 | 86.4 | 73.6 | 37.5 | 100% |
| VTW (AAAI25) [26] | 40% | 9.11 | 53.3 | 1472.8 | 65.6 | 84.1 | 68.3 | 16.3 | 85% |
| PDrop (CVPR25) [46] | 25% | 6.77 | 60.4 | 1462.6 | 65.1 | 86.4 | 68.3 | 27.4 | 92% |
| FastV (ECCV24) [5] | 22% | 5.76 | 60.3 | 1469.1 | 64.3 | 85.5 | 68.2 | **32.3** | 94% |
| DivPrune (CVPR25) [1] | 22% | 4.20 | **61.4** | 1467.9 | 65.4 | 86.2 | 67.4 | 26.9 | 92% |
| **BTP**(ours) | 22% | 4.52 | 60.6 | **1490.8** | **65.8** | **86.7** | **68.4** | 30.3 | **94%** |
| **Qwen2.5-VL-7B *** | | | | | | | | | |
| Original | 100% | 5.48 | 60.4 | 1690.8 | 82.5 | 87.4 | 76.7 | 16.1 | 100% |
| VTW (AAAI25) [26] | 40% | 2.38 | 40.2 | 1129.8 | 58.7 | 61.5 | 69.7 | 4.5 | 65% |
| PDrop (CVPR25) [46] | 30% | 1.81 | 49.9 | 1462.5 | 70.6 | 76.8 | 72.6 | 9.58 | 82% |
| FastV (ECCV24) [5] | 30% | 1.79 | 52.6 | 1595.5 | 73.4 | 83.9 | 74.0 | 16.2 | 96% |
| DivPrune (CVPR25) [1] | 25% | 1.34 | 50.1 | 1639.2 | **76.9** | 85.4 | 73.0 | **17.5** | 96% |
| **BTP**(ours) | 25% | 1.67 | **57.2** | **1651.5** | 75.2 | **86.2** | **74.1** | 16.8 | **97%** |

# 5  Experiment

**Baselines and models**   To rigorously assess the generalizability of our proposed image token compression method, we integrate it into several state-of-the-art multimodal large models and conduct extensive experiments on diverse benchmark tasks. Specifically, we evaluate our approach on four representative models: LLaVA-v1.5-7B, LLaVA-v1.5-13B, LLaVA-v1.6-7B and Qwen2.5-VL-7B-Instruct [2, 27, 28, 29, 43]. We select several plug-and-play compression baselines that support inference-time token pruning: FastV [5] and PyramidDrop [46], which select informative tokens via attention mechanisms; DivPrune [1], which filters tokens based on visual diversity and VTW [26], which discards all image tokens at a specific transformer layer determined by validation performance.

**Benchmarks and evaluation**   We conduct comprehensive experiments on standard visual under-standing tasks using models of different sizes, model families, and compression ratios. We report the results on GQA, MMB, MME, POPE, SQA and MM-VeT [13, 21, 22, 30, 49, 50]. All experiments are carried out using the LMMs-Eval [3, 24] framework. In addition to accuracy on each dataset, we evaluate all methods in terms of FLOPs, inference latency, and KV cache memory usage. For inference throughout, we follow the PyramidDrop. Specifically, we calculate the FLOPs of the $l$-th layer's attention and MLP modules through $4nd^2 + 2n^2d + 3ndm$. $n$ is the number of tokens, $d$ is the hidden state size, and $m$ is the intermediate size of the FFN.

**Implementation details**   All pruning experiments are conducted on 8 NVIDIA A800 GPUs using the HuggingFace Transformers library. To determine pruning stages, we randomly sample 64 instances from the LLaVA-655k [27, 28, 29] dataset and use the same set across all models and

benchmarks, thus avoiding separate calibration for each benchmark. We gradually reduce the number of image tokens at each stage. In the early layers, we use a larger $\lambda$ value to focus more on global information, while in the deeper layers, we use a smaller lambda to emphasize local details. More implementation details for different models are provided in the see Appendix 7.3. Similar to the implementation of PyramidDrop, we compute the required attention scores separately within the **FlashAttn** module at the specified pruning layers, achieving full compatibility with **FlashAttn** [11, 12]. It is worth noting that all our experiments are conducted with **FlashAttention** acceleration enabled.

## 5.1 Main results

**BTP outperforms SOTA methods across LVLMs**    As shown in Table 1, we conduct extensive experiments across different model families and parameter scales. Empirical results demonstrate that our approach consistently surpasses state-of-the-art methods on most benchmark tasks. Our method achieves **98%** of the original average performance under a **22%** compression rate across LLaVA models of different sizes. Moreover, our method consistently outperforms all models, achieving better results than both attention-based and diversity-based approaches. We also visualize the impact of different methods on layer outputs in Figure 5, our method preserves consistency with the original outputs at both local and global levels. The Appendix 7.5 further provides visualizations of the spatial distribution of image tokens selected by various methods. Our method yields more effective token selection in deeper layers.

**BTP maintains stable performance across different compression ratios**    We assess the performance of our method across a range of compression ratios to verify its effectiveness. We find that FLOPs account only for the computational cost of the attention and MLP modules, while ignoring the overhead introduced by additional components. As a result, FLOPs alone fail to accurately reflect the actual inference latency. Therefore, as shown in Table 2, we compare the performance and average inference time of different methods under varying compression ratios. In can be observed that although DivPrune achieves lower theoretical FLOPs, its end to end latency even exceeds that of the original uncompressed model. In contrast,

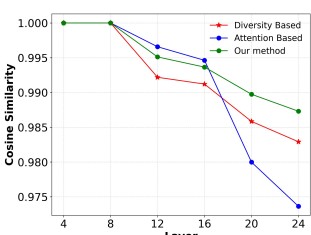

Figure 5: Effect of various pruned methods on the output of decoder layers.

our method leverages spatial division for initialization, significantly reducing the actual inference time. Across various compression ratios, our method consistently achieves better performance than state-of-the-art approaches on most datasets, without incurring additional computational overhead.

Table 2: Performance comparison with FastV and DivPrune across varying compression ratios. We report the results on LLaVa-v1.5-7B.

| Method | Average Token | TFLOPS | Latency | GQA | MME | MMB | SQA |
|---|---|---|---|---|---|---|---|
| LLaVA-1.5-7B | 576 | 3.82 | 0.145s | 62.0 | 1510.7 | 64.3 | 69.4 |
| FastV | 128 | 0.86 | 0.122s(15% ↓) | 49.6 | 1388.6 | 56.1 | 60.2 |
| DivPrune | 128 | 0.83 | 0.224s(54% ↑) | 58.8 | 1405.4 | 62.1 | 68.4 |
| **BTP**(ours) | 128 | 0.85 | 0.134s(7% ↓) | **59.0** | **1487.0** | **62.7** | **69.1** |
| FastV | 64 | 0.42 | 0.118s (18% ↓) | 46.1 | 801.3 | 48.0 | 51.1 |
| DivPrune | 64 | 0.41 | 0.150s(0.5% ↑) | 57.5 | 1350.0 | 58.5 | 67.6 |
| **BTP**(ours) | 64 | 0.42 | 0.120s(17% ↓) | 55.0 | **1364.1** | **58.6** | **68.3** |

## 5.2 Efficiency analysis

The additional overhead introduced by our method primarily arises from the attention computation and the selection of the diversity set. Since we compute attention only between the final token and the image tokens, the added attention complexity is $\mathcal{O}(n)$. For the selection of the diversity set, our proposed spatial initialization strategy and progressive weight decay allow us to select only a small number of additional tokens. In this section, we compare the efficiency of our method with other

approaches, evaluating from multiple perspectives including theoretical FLOPs, inference latency, KV cache size, and corresponding benchmark performance. For inference latency, we report the average inference time per sample. For KV cache memory usage, we report the average GPU memory consumption after compression. We conduct experiments using LLaVA-v1.5 and LLaVA-v1.6. Notably, LLaVA-v1.6 processes images at a higher resolution, resulting in a larger number of image tokens.

Table 3: Evaluation of compression efficiency on different models

| Method | Averge token | Cache Size | TFLOPS | Latency | LLaVA-COCO |
|---|---|---|---|---|---|
| **LLaVA-1.5-7B** | 576 | 0.34GB (100%) | 3.82 | 2.24s | 90.8 |
| FastV | 172 | 0.15GB (55.8% ↓) | 1.65 | 2.11s (5% ↓) | 80.6 |
| DivPrune | 128 | 0.11GB (67.6% ↓) | 0.83 | 2.33s (4% ↑) | 80.3 |
| **BTP**(ours) | 128 | 0.11GB (67.6% ↓) | 0.85 | 2.13s (4% ↓) | **80.9** |
| **LLaVA-1.6-7B** | 2880 | 1.11GB(100%) | 20.82 | 4.24s | 106.6 |
| FastV | 860 | 0.37GB (66.6% ↓) | 6.45 | 3.77s (11%↓) | 92.6 |
| DivPrune | 633 | 0.28GB (74.7% ↓) | 4.20 | 5.00s (17%↑) | **99.1** |
| **BTP**(ours) | 633 | 0.28GB (74.7% ↓) | 4.52 | 3.91s(7%↓) | 98.9 |

As shown in Table 3, our method achieves the best performance while maintaining practical efficiency.

## 5.3   Ablation study

**Choice of balance factor value:** We first analyze the effect of $\lambda$ in the local-global objective functions. This factor determines the trade-off at each layer between preserving local outputs and contributing to the global output. To thoroughly analyze the contribution of each pruning layer, we perform comprehensive ablation experiments on the LLaVA model. Our method includes three pruning layers, and we evaluate three configurations by fixing the $\lambda$ parameters of two layers while varying the remaining one: (1) tuning the shallow layer while fixing the middle and deep layers, (2) tuning the middle layer while fixing the shallow and deep layers, and (3) tuning the deep layer while fixing the shallow and middle layers. We define the ratio between the performance of the pruned model and that of the base model on the target task as the performance gain. The computation of performace performance gain is detailed in the Appendix 7.4.

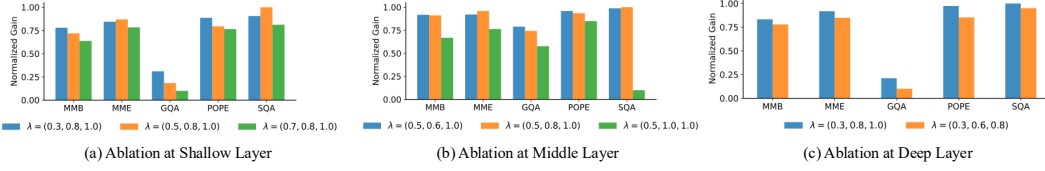

Figure 6: Ablation study on balance factor.

As shown in Figure 6, we can observe an early preference for the diversity objective in the shallow layers results in performance degradation. The middle layers should still retain a moderate degree of diversity, whereas the deeper layers, due to the limited number of remaining tokens, should prioritize the attention objective. This highlights the importance of our method in effectively balancing the two objectives.

**Effectiveness of rebalanced attention and spatial diversity initialization:** We then perform ablation studies on the attention rebalance module and the spatial initialization module. We experimented with various combinations of the two modules. The results are presented in Table 4. It can be observed that removing the attention rebalance module results in a significant degradation in model performance. This degradation arises from the inherent bias in attention mechanisms, where positional encodings tend to shift attention disproportionately toward later tokens, leading to suboptimal token selection. On the other hand, omitting

Table 4: Ablation study on attention rebalance module and spatial initialization module.

| RA | SI | Latency | MME | GQA | POPE |
|---|---|---|---|---|---|
| ✓ | ✓ | 0.134s | **1487.0** | **59.0** | **85.6** |
| ✓ |  | 0.232s | 1486.5 | 57.9 | 86.4 |
|  | ✓ | 0.140s | 1464.6 | 57.4 | 85.1 |
|  |  | 0.231s | 1478.1 | 57.3 | 84.4 |

the spatial initialization module causes a marked increase in inference latency, in some cases even surpassing that of the original unpruned model. This suggests that while pruning reduces token count, naive initialization can introduce computational overhead that negates the benefits of pruning, thereby limiting the method's applicability in latency-sensitive real-world scenarios [55]. This demonstrates the effectiveness of the proposed module in improving both model performance and inference speed. We also conducted an ablation study on the distance definitions used in the spatial diversity initialization module. As shown in the Appendix 7.7, we found that the Euclidean distance models the diversity of image tokens more effectively than the Manhattan distance.

**Effectiveness of calibration-based pruning stage selection:** To evaluate the effectiveness of our proposed calibration-based pruning stage selection, we compare it with a baseline that uniformly divides the pruning stages according to the total number of decoder layers, under the same compression rate. Experimental results are shown in Table 5. We observe that our pruning layer selection method outperforms uniform selection. This is especially evident on Qwen2.5-VL, where uniform selection leads to a significant performance drop. We attribute this to differences in how Qwen2.5-VL processes image tokens as shown in Figure 4. We also conduct an ablation study on the size and composition of the calibration set. Specifically, we expanded the calibration set by incorporating images from multiple datasets, including GQA, $V^*$Bench [45], and SQA and UrbanVideo-Bench [58]. We then repeated the experiment shown in Figure 4 using calibration set

Table 5: Ablation study on layer selection strategy.

| Method | Stage Selection | MME | MMB |
|---|---|---|---|
| LLaVa-v1.5 | Averaged | 1483.2 | 62.3 |
| | Ours | **1487.0** | **62.7** |
| LLaVa-v1.6 | Averaged | 1480.1 | 64.7 |
| | Ours | **1490.8** | **65.8** |
| Qwen2.5-vl | Averaged | 1551.6 | 73.8 |
| | Ours | **1641.5** | **75.2** |

sizes of 64, 128, and 256. The results are presented in Appendix 7.6, we can see that the variation patterns of image tokens remain consistent across different calibration set sizes and content, demonstrating the robustness of our pruning layer selection method.

**Ablation on the Computation of Attention-Based Importance Scores:** In our method, the importance score of each image token is obtained by using the attention assigned to it by the last text token in the prefilling stage. To verify the robustness of this design, we conduct an ablation study comparing different ways of computing the importance score: 1. Averaging attention weights from all text tokens to each image token. 2. Following the approach in [57], where image–text similarity is first computed and the most similar text tokens are then selected to calculate the importance score.

Table 6: Ablation study on importance score calculation method.

| Method | MME | MMB | POPE | GQA | SQA |
|---|---|---|---|---|---|
| last-token(ours) | **1497** | **63.4** | **85.6** | **59.1** | 69.1 |
| averaged-tokens | 1490 | 62.8 | 84.7 | 57.3 | 69.4 |
| similarity-based | 1485 | 63.1 | 84.7 | 57.9 | **69.7** |

The results are shown in Table 6. We can see that last token efficiently modeling the importance score. We believe that the last token in the input prompt is a suitable choice for computing the importance score because it is typically decoded as the first output token during the decoding stage. This allows it to effectively capture the model's focus.

# 6   Conclusion

In this work, we conduct initial studies to investigate and verify the limitations of existing image token pruning methods. We further analyze the impact of two pruning strategies on model performance from the perspective of the objective function, and formulate a local-global pruning optimization objective. To reduce information loss during pruning, we propose **Balanced Token Pruning (BTP)**, a multi-stage pruning method. We first determine the pruning stages using a calibration set. In the early layers, we focus on a *diversity-oriented objective* to account for the influence of pruning on deeper layers, while in the later layers, we adopt an *attention-based objective* to better preserve local information. In future work, we will further investigate the lightweight deployment on real devices [61] and explore its potential applications in multi-agent collaboration [36, 16].

## Acknowledgments

This paper was supported by the Natural Science Foundation of China under Grant 62371269 and 62272262, Shenzhen Low-Altitude Airspace Strategic Program Portfolio Z253061, Guangdong Innovative, Entrepreneurial Research Team Program (2021ZT09L197), Meituan Academy of Robotics Shenzhen and Tsinghua University-Toyota Research Center.

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

# 7 Appendix

## 7.1 Key and Value of LVLMs

Following previous works on token quantization KIVI [31], we visualize the $K_l$ and $V_l$ of different LVLMs, the results are shown below:

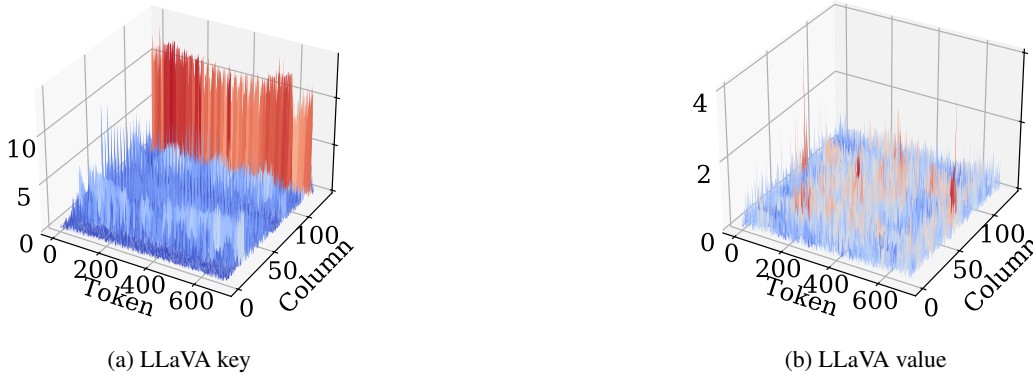

(a) LLaVA key                                      (b) LLaVA value

Figure 7: Visualization of key and value of LLaVA-v1.5

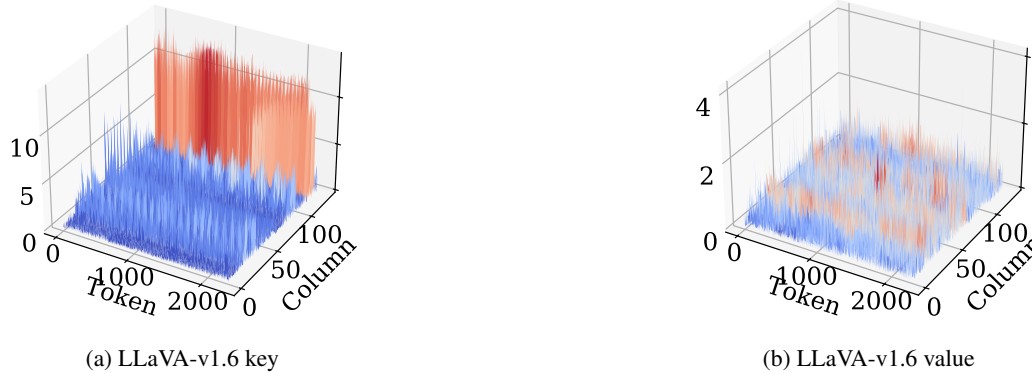

(a) LLaVA-v1.6 key                                 (b) LLaVA-v1.6 value

Figure 8: Visualization of key and value of LLaVA-v1.6

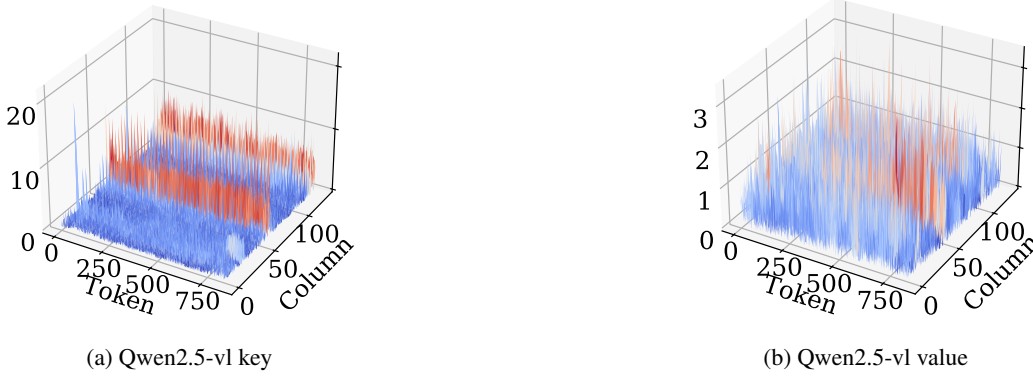

(a) Qwen2.5-vl key                                 (b) Qwen2.5-vl value

Figure 9: Visualization of key and value of Qwen2.5-vl

## 7.2 Top-k Importance Image Token Received Attention Ratio

We calculate the ratio between the attention scores received by the top-k most text-attended image tokens and the total attention scores received by all image tokens:

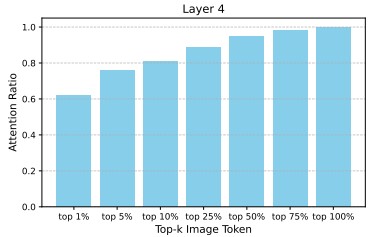

(a) Layer 4 Attention Ratio

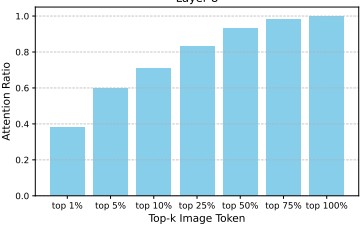

(b) Layer 8 Attention Ratio

Figure 10: Visualization Top-k Importance Image Token Received Attention Ratio

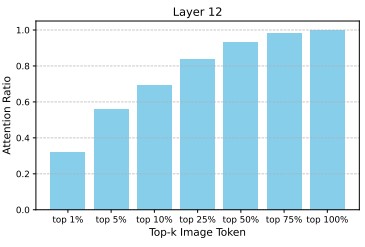

(a) Layer 12 Attention Ratio

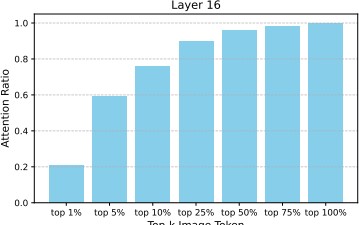

(b) Layer 16 Attention Ratio

Figure 11: Visualization Top-k Importance Image Token Received Attention Ratio

## 7.3 Experiment Settings

For `LLaVA-v1.5-7B`, `LLaVA-v1.5-13B`, and `LLaVA-v1.6-7B`, we divide the pruning process into five stages based on the image token handling pipeline described in the Appendix. In each stage, except for the last one, we retain 50% of the tokens from the previous stage. In the final stage, all tokens are discarded to maximize inference speed. For `Qwen2.5-VL`, since its image token processing can be clearly divided into two stages, we retain 25% of the tokens in the fourth stage and 12.5% in the final stage to preserve model performance. The $\lambda$ used for different models are shown below:

Table 7: $\lambda$ settings in different models

| Model | $\lambda$ |
|---|---|
| llava-v1.5-7b | (0.6,0.8,1.0) |
| llava-v1.5-13b | (0.6,0.8,1.0) |
| llava-v1.6-13b | (0.4,0.7,1.0) |
| qwen-2.5-vl-7b | (0.2,0.5,0.8,1.0) |

## 7.4 Calculation of model gain

Since evaluation metrics vary across tasks and the difficulty levels differ significantly, it is not reasonable to present all task results directly in a unified format. For example, the original LLaVA-v1.5 model scores 1510 on the MME benchmark but only 62 on GQA. To address this, we define a model gain metric as:

$$Gain = Normalize(\frac{Pruned_{score}}{Original_{score}}). \tag{12}$$

## 7.5 Visualization of token selection under different pruning strategies

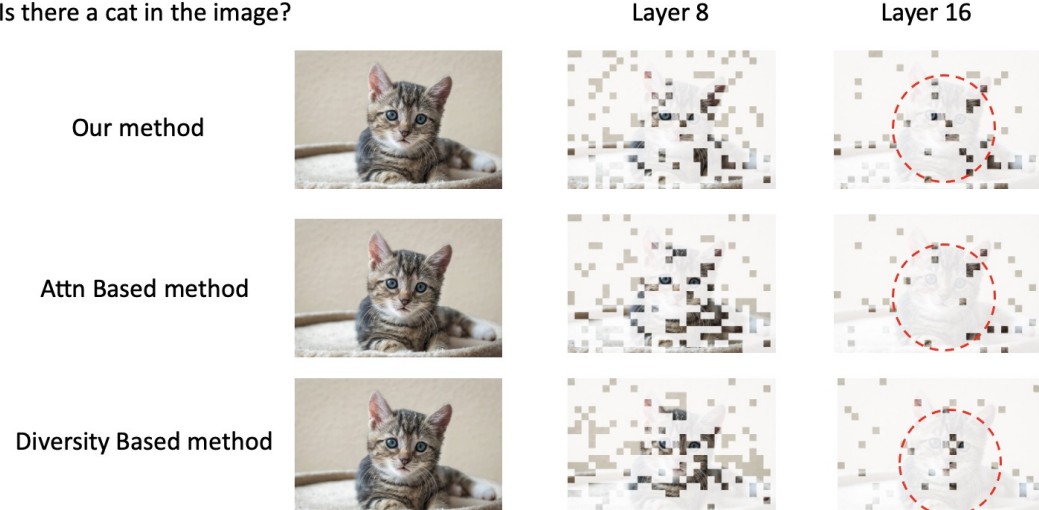

Figure 12: Visualization of Image Token Selection Across Different Methods

## 7.6 Ablation Study on Calibration Set

According to Equation 2 in the paper, in multimodal large language models, M denotes the causal mask, which constrains each token to attend only to preceding tokens. In our input format, image tokens always precede the text prompts (i.e., the input follows the structure: prefix + image + question). As a result, the model processes the image tokens before it receives the specific question which is unrelated to the input question. To validate our hypothesis, we posed different types of questions on the same image. We then conducted the experiment presented in Figure 4 using LLaVA-v1.5. We computed the number of image tokens whose cosine similarity between adjacent layers falls below 0.93. The resulting trends are shown as follows:

Table 8: Ablation on Calibration Set Size.

| Set Size | layer1 | layer5 | layer9 | layer13 | layer17 | layer21 | layer25 |
|----------|--------|--------|--------|---------|---------|---------|---------|
| 64 | 0 | 0 | 325 | 141 | 155 | 45 | 1 |
| 128 | 0 | 0 | 325 | 141 | 155 | 45 | 1 |
| 256 | 0 | 0 | 325 | 141 | 155 | 45 | 1 |

We observe that varying the question type for the same image does not lead to significant differences in the results. In the following analysis, we investigate how image content and the size of the calibration set affect our method. We then repeated the experiment shown in Figure 4 using calibration set sizes of 64, 128, and 256. Specifically, we computed the number of image tokens whose cosine similarity between adjacent layers falls below 0.93 using LLaVA-v1.5. The results are presented below:

Table 9: Ablation on Calibration Set Size.

| Set Size | layer1 | layer5 | layer9 | layer13 | layer17 | layer21 | layer25 |
|----------|--------|--------|--------|---------|---------|---------|---------|
| 64 | 0 | 0 | 325 | 141 | 155 | 45 | 1 |
| 128 | 0 | 0 | 350 | 166 | 127 | 36 | 5 |
| 256 | 0 | 0 | 332 | 174 | 145 | 32 | 3 |

## 7.7 Ablation Study on Distance

To evaluate the impact of this choice, we conducted an ablation study comparing two different distance metrics using llava-v1.5. The results are summarized in the table below:

Table 10: Ablation Study on Distance

| Method | MME | MMB | POPE | GQA | SQA |
|---|---|---|---|---|---|
| Manhattan | 1497.0 | 63.4 | **85.6** | **59.1** | 69.1 |
| Euclidean | **1506.0** | **64.0** | **85.6** | 58.9 | **69.3** |

