# OpenReview forum: "Balanced Token Pruning: Accelerating Vision Language Models Beyond Local Optimization"
_NeurIPS.cc/2025/Conference — NeurIPS 2025 poster_

### Official Review · Reviewer_jfqF · 2025-06-27

**Clarity:** 2
**Significance:** 2
**Originality:** 3
**Rating:** 4
**Confidence:** 2

**Summary:**

This paper proposes a novel Balanced Token Pruning (BTP) method to reduce the computational cost of large Vision-Language models. Through empirical analysis, the authors show that attention-based pruning mainly focuses on locality, while diversity-based pruning considers the impact on subsequent layers. BTP combines the strengths of both approaches by applying a diversity-based objective in the early layers to select semantically diverse tokens, and an attention-based objective in the later layers to locally optimize each layer and minimize output error. To support this framework, the paper introduces an attention rebalance module and a spatial initialization strategy. It also proposes a layer selection mechanism using a small calibration image set. Experiments applying the proposed method to various models (LLaVA-1.5-7B, LLaVA-1.5-13B, LLaVA-1.6-7B, and Qwen2.5-VL-7B) show that BTP achieves better performance not only compared to attention-based pruning, but also to diversity-based pruning methods. Ablation studies further validate the effectiveness of each component.

**Questions:**

**1. Spatial Initialization**

Could you explain why you chose Manhattan distance instead of Euclidean distance? For example, given a source token at (2, 2), both (6, 6) and (2, 10) have the same Manhattan distance, but different Euclidean distances. In many spatial contexts, Euclidean distance may provide a more accurate measure of proximity.


**2. Pruning Layer Selection**

In Qwen2.5-VL, significant changes in image token representations are observed in layers 10 and 24, yet relatively fewer image tokens are selected in those layers. This appears inconsistent with the assumption that image tokens receive more attention in layers with significant feature changes. LLaVA-v1.5 shows a similar pattern—although layers 8–12 exhibit large representational shifts, more image tokens are selected in layers 17, 19, and 20. Further clarification would be helpful.

**3. Attention Optimization**

The method uses the last token in the input prompt ($X_T$) to compute importance scores. It is unclear whether relying on a single token is sufficient to capture the full contextual relevance of the prompt. Further explanation of this choice would improve the understanding of its effectiveness.

**4. $\lambda_i$ Values**

The $\lambda_i$ values are critical hyperparameters that influence the performance of the proposed framework. In Section 5.3, the authors apply smaller lambda values (0.5–0.7) to shallow layers and larger ones (0.8–1.0) to deeper layers. However, it is not clearly explained how the $\lambda_i$ values are determined for each layer. It would be beneficial to provide a more concrete rule or conduct an ablation study on the impact of $\lambda_i$ settings.

**5. Inference Latency and Spatial Initialization**
In Table 4, when the attention rebalance (RA) module is used, adding spatial initialization significantly improves inference latency. However, in cases where RA is not used, the latency difference with or without spatial initialization becomes marginal. An explanation for this discrepancy would be appreciated.


**6. Appendix A.1**

In Section 4.1, the authors present the assumption that when $V_l$ values are similar, selecting image tokens with high importance scores can effectively reduce the difference in outputs before and after pruning. However, the only supporting evidence is a figure, and no explanation is provided. This lack of explanation makes the assumption difficult to evaluate.


**7. Equation (1)**

The equation for $\text{Atten}^{(l)}$ is not explicitly defined in the paper. Including a clear mathematical formulation would help readers better understand how attention scores are computed at each layer.

**Ethical Concerns:**

["NO or VERY MINOR ethics concerns only"]

**Final Justification:**

While I greatly appreciate efforts and the additional insights provided, I believe that my current score remains appropriate based on my overall assessment of the work, and I have therefore chosen to maintain it.

**Limitations:**

yes

**Paper Formatting Concerns:**

There are no formatting issues.

**Quality:**

2

**Strengths And Weaknesses:**

***Strenghts***

1. The paper is well-motivated, supported by empirical analysis comparing attention-based and diversity-based pruning.
2. The proposed method demonstrates superior performance over both attention-based and diversity-based pruning across multiple large-scale VL models.

***Weaknesses***

1. Some experimental results are missing or not fully detailed, limiting the understanding of the method’s generalizability.
2. Certain components lack sufficient explanation.

---

> ### Author Rebuttal · Authors · 2025-07-30
>
> Thank you for your insightful feedback. We appreciate the opportunity to address the concerns you raised. Below, we provide our responses to the problems:
>
> ---
>
> **Q1**: Spatial Initialization: Could you explain why you chose Manhattan distance instead of Euclidean distance?
> **R1**: Thank you for your constructive feedback. We did not apply any special design when choosing the distance metric. To evaluate the impact of this choice, we conducted an ablation study comparing two different distance metrics using llava-v1.5. The results are summarized in the table below:
> | Method | MME | MMB | POPE | GQA | SQA |
> |----|:----:|:----:|:----:|:----:|:----:|
> | Manhattan | 1497 | 63.4 | **85.6** | **59.1** | 69.1 |
> | Euclidean | **1506** | **64.0** | **85.6** | 58.9 | **69.3** |
>
> We found that Euclidean distance indeed performs better for spatial initialization. We will incorporate this finding into Section 5.3 of the revised paper and update our implementation to adopt Euclidean distance in the next version.
>
> ---
>
> **Q2**: Pruning Layer Selection: In Qwen2.5-VL, significant changes in image token representations are observed in layers 10 and 24, yet relatively fewer image tokens are selected in those layers. This appears inconsistent with the assumption that image tokens receive more attention in layers with significant feature changes. LLaVA-v1.5 shows a similar pattern—although layers 8–12 exhibit large representational shifts, more image tokens are selected in layers 17, 19, and 20. Further clarification would be helpful.
> **R2**: Thank you very much for your question—we appreciate the opportunity to clarify this point. Firstly, once the pruning layers are determined, our method retains more tokens in earlier layers than in deeper ones. As described in Section 4.2, the token set preserved in deeper layers is a subset of that from earlier layers. This design is consistent with prior work [1], and therefore, it is not the case that more image tokens are retained in deeper layers.
>
> Secondly, our hypothesis in Section 4.3 is: *"LVLMs tend to allocate more attention to image tokens in layers following those where the representations of image tokens **undergo** significant changes,"* which is different from *"image tokens receive more attention in layers with significant feature changes."*
>
> To illustrate, we only prune a selected subset of layers. The tokens retained in a pruned layer are propagated through subsequent layers. Our method favors diversity in early layers and attention-based selection in deeper layers. For example, in LLaVA-v1.5, we retain more tokens in layers 8–12 and fewer in layers 17, 19, and 20. If we have misunderstood your concern, we would be happy to engage in further discussion.
>
> ---
>
> **Q3**: Attention Optimization: The method uses the last token in the input prompt $X_T$ to compute importance scores. It is unclear whether relying on a single token is sufficient to capture the full contextual relevance of the prompt. Further explanation of this choice would improve the understanding of its effectiveness.
> **R3**: Thank you very much for your helpful suggestion. We agree that further exploring the computation of the importance score is valuable.
>
> To this end, we compared with two more approaches: (1) computing the importance score by averaging over all text tokens following the image tokens, and (2) the method used in [2], which first computes image-text similarity and then selects the most similar text tokens to compute the importance score. The results are shown in the table below:
> | Method | MME | MMB | POPE | GQA | SQA |
> |----|:----:|:----:|:----:|:----:|:----:|
> | last-token(ours) | **1497** | **63.4** | **85.6** | **59.1** | 69.1 |
> | averaged-tokens | 1490 | 62.8 | 84.7 | 57.3 | 69.4 |
> | similarity-based | 1485 | 63.1 | 84.7 | 57.9 | **69.7** |
>
> We believe that the last token in the input prompt is a suitable choice for computing the importance score because it is typically decoded as the first output token during the decoding stage. This allows it to effectively capture the model's focus. We will include this discussion in the ablation study section of the revised paper.
>
> ---
>
> **Q4**: The values $\lambda_i$ are critical hyperparameters that influence the performance of the proposed framework. It would be beneficial to provide a more concrete rule or conduct an ablation study on the impact of $\lambda_i$ settings.
> **R4**: Thank you for your constructive comment. In our experiments, the $\lambda_i$ parameter was manually selected. For a given model, the $\lambda_i$ value remains fixed across different benchmarks. We only set $\lambda_i$ values at pruning layers. The table below summarizes the $\lambda_i$ values used for each model at pruning layers:
> | Model | $\lambda$ of different layers |
> |:------:|:------:|
> |   llava-v1.5-7b   |   (0.6,0.8,1.0)  |
> |   llava-v1.5-13b   |   (0.6,0.8,1.0)   |
> |   llava-v1.6-7b  |   (0.4,0.7,1.0)   |
> |   Qwen-2.5vl-7b   |   (0.4,0.6,0.8,1.0)   |
>
> Once the pruned layers are determined, the $\lambda$ parameter is also fixed. To further investigate the sensitivity of our method to the choice of $\lambda$, we conduct an ablation study on LLaVA-v1.5-7B by varying $\lambda$ at different stages. The results are shown below:
> **Experiments in the shallow layer**
> | $\lambda$ | MMB| MME | GQA | POPE| SQA |
> |------|:------:|:------:|:------:|:------:|:------:|
> |   (0.3,0.8,1.0)   |   **63.4**   |   1497   |   **58.9**   |  **85.3**    |   69.1   |
> |   (0.5,0.8,1.0)   |   63.1   |   **1500**   |   58.3   |  84.7    |    **69.6**  |
> |   (0.7,0.8,1.0)   |  _62.7_    |   _1490_   |   _57.9_   |  _84.5_    |   _68.6_   |
>
> **Experiments in the middle layer**
> | $\lambda$ | MMB| MME | GQA | POPE| SQA |
> |------|:------:|:------:|:------:|:------:|:------:|
> |   (0.5,0.6,1.0)   |   **63.2**   |   1486   |   **59.0**   |   **85.2**   |   **69.4**   |
> |   (0.5,0.8,1.0)   |   63.1   |   **1500**   |   58.3   |  84.7    |    69.6  |
> |   (0.5,1.0,1.0)   |   _59.3_   |   _1428_  |  _55.8_    |   _82.9_   |  _54.4_    |
>
> **Experiments in the deep layer**
> | $\lambda$ | MMB| MME | GQA | POPE| SQA |
> |------|:------:|:------:|:------:|:------:|:------:|
> |   (0.3,0.8,1.0)   |   **63.4**   |   **1497**   |   **58.9**   |  **85.3**    |   **69.1**   |
> |   (0.3,0.6,0.8)   |   63.2   |   1491   |   58.5   |  84.3    |    68.9  |
>
> We found that using a smaller $\lambda$ value in earlier layers encourages the pruning strategy to focus more on the global objective, while employing a larger $\lambda$ value in deeper layers helps preserve local outputs. This observation further supports the effectiveness of our proposed method. We will include this discussion in the ablation study section of the revised paper.
>
> ---
>
> **Q5**: Inference Latency and Spatial Initialization In Table 4, when the attention rebalance (RA) module is used, adding spatial initialization significantly improves inference latency. However, in cases where RA is not used, the latency difference with or without spatial initialization becomes marginal. An explanation for this discrepancy would be appreciated.
> **R5**: We sincerely apologize for the confusion caused by the writing error. After reviewing our experiment logs and re-running the experiment across three datasets, we found that the averaged latency in the last row of Table 4 should be **0.231s**. We will correct this in the revised version of the paper. Thank you again for your careful review—your feedback has been very helpful to us.
>
> ---
>
> **Q6**: In Section 4.1, the authors present the assumption that when values are similar, selecting image tokens with high importance scores can effectively reduce the difference in outputs before and after pruning. This lack of explanation makes the assumption difficult to evaluate.
> **R6**: We appreciate the opportunity to clarify this point. As shown in Equation (1), the hidden states at each pruned layer are primarily influenced by the outputs of both the attention and MLP modules. Since the MLP parameters remain fixed and its inputs include the outputs from the attention module, we attribute the majority of the observed changes in the hidden states to variations in the output of the attention module.
>
> Further, according to Equation (2), the attention output is determined by the attention scores and the value matrix $V_l$. As detailed in Appendix A.1, we observe that the value vectors across different image tokens exhibit similar magnitudes. Therefore, maximizing the sum of attention scores effectively enhances the similarity of local outputs, which aligns with the assumption.
>
> ---
>
> **Q7**: The equation for $Attn^l$ is not explicitly defined in the paper. Including a clear mathematical formulation would help readers better understand how attention scores are computed at each layer.
> **R7**: Thank you very much for your suggestion. We denote $\mathrm{Attn}^l$ as the attention module at layer $l$, with its formulation defined as:
>
> $$\mathrm{Attn}^l = softmax({\frac {Q_lK_l^T+M}{\sqrt d_k}V_l}).$$
>
> where M is the casual mask. We will include this definition in Section 3 Preliminaries of the next version of the paper.
>
> [1] PyramidDrop: Accelerating Your Large Vision-Language Models via Pyramid Visual Redundancy Reduction. CVPR2025.
> [2] SparseVLM: Visual Token Sparsification for Efficient Vision-Language Model Inference. ICML2025.

---

> > ### Comment · Reviewer_jfqF · 2025-08-05
> >
> > Thank you very much for a detailed and thoughtful rebuttal.
> > Your comments greatly helped me understand the intuition and design choices of your method.
> >
> > However, I still have some concern regarding the selection of the hyperparameter $\lambda_{i}$.
> > In the additional experiments provided in your rebuttal, the results indicate that smaller values of $\lambda_{i}$ in shallow layer lead to better performance, while larger values of $\lambda_{i}$ in deep layer perform better.
> > I fully agree that this observation supports your method's intuition.
> >
> > But, the actual values of $\lambda_{i}$ applied in your main experiments (for example, (0.6, 0.8, 1.0) for LLaVA-v1.5-7B) seem somewhat inconsistent with the trends observed in the ablation results.
> > Based on the experimental findings, a configuration such as (0.3, 0.6, 1.0) might potentially yield better results.
> >
> > Could you clarify why (0.6, 0.8, 1.0) was chosen despite the experimental evidence suggesting otherwise?
> > Was the selection made for reasons such as stability, consistency across models, or other practical considerations?
> >
> > Thank you again for your thoughtful response and engagement.

---

> > > ### Author Response · Authors · 2025-08-05
> > >
> > > We sincerely thank you for your timely and thoughtful feedback. We are committed to addressing your concerns to the best of our ability.
> > >
> > > ---
> > >
> > > **Q1**: Could you clarify why (0.6, 0.8, 1.0) was chosen despite the experimental evidence suggesting otherwise? Was the selection made for reasons such as stability, consistency across models, or other practical considerations?
> > >
> > > **R1**: Thank you for your constructive feedback. Regarding the selection of $\lambda_i$, we acknowledge that we did not conduct an extensive search over the hyper-parameter space during the initial experiment setup. Instead, we adopted a heuristic approach: starting from the final layer, where the attention allocation was assigned a weight of 1, we chose the hyper-parameters based on an arithmetic progression. (for example, (0.6, 0.8, 1.0) for LLaVA-v1.5-7B，(0.4,0.6,0.8,1.0) for Qwen-2.5vl-7B).
> > >
> > > Your observation is absolutely correct. Below, we provide a comparison between the hyperparameters used in Table 1 of our paper (0.6,0.8,1.0) and those used in the ablation study (0.3,0.8,1.0):
> > > | $\lambda$ | MMB | MME | GQA | POPE | SQA |
> > > |------|:------:|:------:|:------:|:------:|:------:|
> > > | (0.6,0.8,1.0)    | 62.7   | 1487   | **59.0**   | **85.6**   | 69.1   |
> > > | (0.3,0.8,1.0)   | **63.4**(+0.7)   | **1497**(+10)   | 58.9(-0.1)   | 85.3(-0.3)   | 69.1   |
> > >
> > > As shown, selecting a smaller $\lambda$ in the shallower layers leads to a more significant performance improvement. Moreover, since more image tokens are retained in the shallower layers, the tokens selected by the attention-based method tend to partially overlap with those selected by the diversity-based method. As a result, only substantial changes in the value of $\lambda$ lead to noticeable performance differences, while minor variations have limited impact on the experimental outcomes. We believe this observation does not affect the validity of our experimental conclusions.
> > >
> > > If you have any further questions, we would be more than happy to engage in additional discussion.

---

> > > > ### Comment · Reviewer_jfqF · 2025-08-05
> > > >
> > > > Thank you very much for your thoughtful response.
> > > >
> > > > Your response has addressed most of my concerns, and I now have a much better understanding of your design choices and results.
> > > >
> > > > While I greatly appreciate your efforts and the additional insights provided, I believe that my current score remains appropriate based on my overall assessment of the work, and I have therefore chosen to maintain it.
> > > >
> > > > Thank you again for your efforts and engagement.

---

> > > > > ### Author Response · Authors · 2025-08-05
> > > > >
> > > > > We are grateful for your constructive engagement during the discussion phase, which we believe will contribute meaningfully to enhancing the quality of our paper. Thank you once again.

---

### Official Review · Reviewer_wCTc · 2025-07-03

**Clarity:** 3
**Significance:** 4
**Originality:** 4
**Rating:** 5
**Confidence:** 4

**Summary:**

This paper introduces Balanced Token Pruning (BTP), a plug-and-play method to reduce the computational cost of  LVLMs by more effectively pruning image tokens. Unlike prior approaches that rely solely on local attention or token diversity, BTP balances both local and global impacts of token pruning across model layers using a multi-stage strategy informed by a small calibration set. Experiments across several LVLMs show that BTP consistently improves efficiency with minimal performance loss.

**Questions:**

- Line 41 - 42: Is this phenomenon of different image tokens being attended by the text tokens consistently observed across the dataset? It seems counter-intuitive to me. At least in vision transformers, we do observe the same tokens being attended to by other tokens throughout the layers.
- Lines 44 - 56: Why was the hidden state of only last word considered for this experiment? What dataset and model was used for this experiment? Overall, I think it would be insightful to add some more explanation (if any) on the observations shown in figure 1 and 2. But this idea of creating a new local-global objective is quite interesting.
- Having different number of tokens pruned for different input sample breaks the regularity in processing batched input. Does it not deteriorate the throughput?

**Ethical Concerns:**

["NO or VERY MINOR ethics concerns only"]

**Final Justification:**

The authors have provided a rebuttal that directly addresses the key suggestion: clarifying CLAdapter's effectiveness across tasks with varying amounts of training data.
Given the technical soundness, novelty, results, and other reviewers’ comments, I would like to raise my rating.

**Limitations:**

yes

**Paper Formatting Concerns:**

No concerns.

**Quality:**

3

**Strengths And Weaknesses:**

- The paper gives a very interesting insight on how token pruning affects the subsequent blocks which is often overlooked in the existing approaches.
- The approach is convincing, in that, a combination of local-global consideration for pruning would be more optimal than the exisiting pruning approaches.
- BTP has been evalulated extensively and compared with many recent token pruning approaches in VLMs.
- Some empirical observations, such as token attention patterns across layers, are counter-intuitive and insufficiently explained, with missing experimental details and unclear generality across the dataset.
- The method’s use of input-dependent token pruning may affect batching efficiency and inference throughput, but this practical concern is not addressed.

---

> ### Author Rebuttal · Authors · 2025-07-30
>
> Thank you for your thorough review and for the positive feedback on our work. We appreciate the opportunity to address your questions. Below, we provide detailed responses to the concerns you raised.
>
> ---
>
> **Q1**: Line 41 - 42: Is this phenomenon of different image tokens being attended by the text tokens consistently observed across the dataset? It seems counter-intuitive to me. At least in vision transformers, we do observe the same tokens being attended to by other tokens throughout the layers.
> **R1**: We are very pleased to have the opportunity to discuss this phenomenon with the reviewer. Yes, we have observed the pattern shown in Figure 1 across different types of datasets, including POPE, GQA, and COCO.
>
> We believe that the reviewer’s observation and our findings are not contradictory. As shown in Figure 1, certain image tokens are indeed attended by multiple layers. Our analysis reveals that the overlap of highly attended image tokens is high between adjacent layers, but this overlap significantly decreases between non-adjacent layers. In the table below, we report the overlap of the top 144 most attended image tokens across different layers:
> | layer1-layer2 | layer2-layer3 | layer8-layer9 | layer10-layer11 | layer16-layer17 | layer18-layer19 | layer24-layer25 | layer26-layer27 |
> |----|----|----|----|----|----|----|----|
> |  0.604  |  0.784  |  0.881  |  0.875  |  0.895  |  0.875  |  0.861  |  0.854  |
>
> | layer1-layer8 | layer2-layer10 | layer8-layer16 | layer10-layer18 | layer16-layer24 | layer18-layer26 |
> |----|----|----|----|----|----|
> |  0.486  |  0.576  |  0.687  |  0.694  |  0.555  |  0.641  |
>
> We observe that the overlap of highly attended image tokens is relatively high between adjacent layers, while it significantly decreases between layers that are farther apart. This suggests that adjacent layers tend to focus on similar regions of the input image, whereas distant layers shift their attention to different regions. Moreover, this trend becomes even more pronounced when computing the overlap using a smaller subset of top-attended image tokens.
>
> ---
>
> **Q2**: Lines 44 - 56: Why was the hidden state of only last word considered for this experiment? What dataset and model was used for this experiment? Overall, I think it would be insightful to add some more explanation (if any) on the observations shown in figure 1 and 2. But this idea of creating a new local-global objective is quite interesting.
> **R2**: We use coco dataset to conduct this experiment with llava-v1.5-7b. We have also observed similar patterns across different datasets and model architectures, suggesting that this phenomenon is consistent and not specific to a particular setting. As shown in Figure 1, diversity-based pruning tends to preserve image tokens that are important for deeper layers, while attention-based pruning primarily retains tokens that are important for the current layer. This discrepancy leads to the results observed in Figure 2.
>
> In Figure 2, our goal is to investigate the impact of pruning during the prefilling stage on the model’s output. Due to the autoregressive nature of large language models, the last token in the prefilling stage is decoded as the first token in the decoding stage. It is also the only “model output” that we can access during the prefilling phase. Therefore, we select the hidden state of the last token in the prefilling stage for analysis.
>
> **Effect of Different Pruning Strategies on the Full Output**: However, in captioning tasks, the model generates multiple tokens in an autoregressive manner. Therefore, we further investigate how output tokens at different positions are affected by different pruning strategies. Specifically, we analyzed the impact on the 1st, 4th, and 20th generated tokens with llava-v1.5-7b. The results are summarized below:
> | Method-Position | Layer4 | Layer8 | Layer12 | Layer16 | Layer20 | Layer24 |
> |------|:------:|:------:|:------:|:------:|:------:|:------:|
> |   Attn(1st)   |   1   |    1  |  **0.996**    |   **0.994**   |  0.979    |   0.973   |
> |   Div(1st)   |    1  |    1  |  0.992    |   0.991   |    **0.985**  |   **0.982**   |
> |   Attn(4th)   |   1   |   **0.998**  |   **0.993**   |   **0.991**   |   0.954   |  0.949    |
> |   Div(4th)   |   1   |   0.995 |    0.989  |    0.989  |   **0.975**   |  **0.972**    |
> |   Attn(15th)   |   **0.369**   |  **0.311**  |  **0.375**   |   0.279   |  0.303 |   0.352   |
> |   Div(15th)  |   0.299   |   0.204   |  0.274   |   **0.312**  |  **0.423**   |    **0.496**  |
> |   Attn(20th)   |   **0.297**   |  0.213  |  0.141   |   0.161   |  0.168 |   0.166   |
> |   Div(20th)  |   0.292   |   **0.255**   |  **0.162**    |   **0.171**   |  **0.169**    |    **0.235**  |
>
> We observe that tokens generated at later positions are more influenced by previously autoregressively generated tokens, resulting in lower similarity to their original hidden states. However, across all positions, the impact trends of the two pruning strategies remain consistent: attention-based pruning preserves local information, while diversity-based pruning better maintains global information.
> **Further Explanation of Figure 1**: We attempt to provide a deeper explanation of the phenomenon shown in Figure 1. Since the multimodal large language model cannot access future questions during image processing due to the causal mask, the image token processing pipeline remains the same across different types of questions. This implies that different layers may encode different aspects of image information. As illustrated in Figure 4, image tokens gradually align to the language space in deeper layers. Moreover, MLLMs tend to assign uniform attention scores to all image tokens in early layers, and only focus on semantically relevant different regions in deeper layers. We plan to further explore this interesting phenomenon in future work.
>
> ---
>
> **Q3**: Having different number of tokens pruned for different input sample breaks the regularity in processing batched input. Does it not deteriorate the throughput?
> **R3**: Thank you for the insightful question. Our method supports batch inference for models that encode images into a fixed number of image tokens. This is because, under fixed pruning layers and pruning ratios, the number of retained tokens before each layer remains consistent across samples, avoiding dimensional mismatches during inference.
>
> However, for models such as LLaVA-v1.6 and Qwen-2.5-VL that employ dynamic resolution and encode different images into varying numbers of image tokens, our approach does pose challenges for batch inference. This presents an interesting trade-off. For example, one potential solution is to determine the number of image tokens to discard based on the sample with the fewest tokens in a batch, thereby ensuring input dimensional consistency across the batch. We plan to explore this further in future work, investigating whether pruning and token reuse strategies can be adapted to support efficient batch inference under such dynamic settings.

---

> ### Comment · Reviewer_wCTc · 2025-08-06
>
> Thanks for the detailed rebuttal. I appreciate the effort you've put into addressing each of the concerns.
>
> On Q1, the additional analysis on token overlap across adjacent vs. distant layers is insightful. The overlap table adds strong empirical support to the explanation.
>
> Regarding Q2, thanks for clarifying the motivation behind using the last token’s hidden state in the prefilling stage. The contrast between attention-based and diversity-based pruning and how they affect local vs. global information is demonstrated and adds depth to your findings.
>
> For Q3, I appreciate the discussion of the trade-offs involved with dynamic token counts during batch processing. It’s good to know that your method still supports batch inference under fixed conditions, and I agree that this is a meaningful direction to explore further.
>
> Overall, the clarifications and additional results make your work stronger. I have no further concerns.

---

> > ### Author Response · Authors · 2025-08-06
> >
> > We are very glad that we have addressed your concerns. Thank you sincerely for your positive feedback and recognition of our work. It was a pleasure discussing with you during the rebuttal phase. We truly appreciate your support and constructive input.

---

> ### Author Response · Authors · 2025-08-07
>
> We’re very glad to hear your positive feedback that 1) our clarifications and additional results have strengthened the paper and 2) you currently have no further concerns.
>
>
> We would be grateful if you could consider raising your score. Thanks!
>
>
> Thank you again for your valuable time and thoughtful feedback!

---

### Official Review · Reviewer_W7Hn · 2025-07-03

**Clarity:** 3
**Significance:** 2
**Originality:** 3
**Rating:** 4
**Confidence:** 3

**Summary:**

This paper aims to address the computational overhead in Large Vision-Language Models (LVLMs) caused by the large number of image tokens. The authors begin by empirically analyzing existing token pruning methods, categorizing them into attention-based and diversity-based approaches. They find that attention-based methods excel at optimizing the output of the current layer (a local objective) but neglect the impact on subsequent layers (a global objective). Conversely, diversity-based methods better preserve global information but sacrifice local output consistency.

To address this local-global trade-off, the authors propose Balanced Token Pruning (BTP), a plug-and-play method for visual token pruning. The core idea of BTP is to divide the pruning process into multiple stages and dynamically adjust the weight between local and global objectives. Specifically, in the early layers of the model, the method emphasizes a diversity-based criterion to preserve rich information that has a significant impact on subsequent layers. In the deeper layers, it shifts focus to an attention-based criterion to ensure the accuracy of the immediate layer's output. Additionally, the method incorporates several supporting techniques: 1) using a small calibration set to automatically select the pruning layers; 2) an attention rebalancing operation to correct for positional biases; and 3) a spatial initialization strategy to accelerate diversity calculations.

**Questions:**

Refer to weaknesses

**Ethical Concerns:**

["NO or VERY MINOR ethics concerns only"]

**Final Justification:**

I thank the authors for their detailed and timely response. After reading the authors' feedback, I believe most of my concerns are addressed, and I decide to keep my original rating, leaning toward acceptance.

**Limitations:**

yes

**Quality:**

3

**Strengths And Weaknesses:**

**Strengths**

1.  The authors clearly point out the inherent contradiction between "local optimization" and "global impact" through an in-depth analysis of existing methods. The visualization analysis of Figures 1 and 2 intuitively shows the difference in behavior of different pruning strategies at different depths of the model, providing a solid foundation and convincing motivation for the "balance" idea proposed in the paper.

2. The core idea of ​​BTP - dynamically balancing local (attention) and global (diversity) goals according to the network depth where the pruning is located - is very novel and intuitive. Rather than simply mixing the two methods, it proposes a principled, staged strategy, that is, protecting information diversity at shallow levels to benefit the global, and focusing on task relevance at deep levels to ensure local. This idea is very elegant.

3. Exhaustive Experimental Validation: The experimental part of the paper is very solid. The author tested on multiple models (LLaVA-v1.5/v1.6, Qwen2.5-VL) and multiple sizes (7B, 13B), covering a wide range of visual understanding benchmarks. The ablation study is very comprehensive and systematically verifies the effectiveness of each component such as balancing factor, rebalancing module, spatial initialization and layering strategy.

**Weaknesses**
1. The paper makes completely unclear statements when describing how the core balance factor $\lambda$ changes with the depth of the network.
- In Section 5.3, the author describes: "In the early layers, we use a larger $\lambda$ value to focus more on global information, while in the deeper layers, we use a smaller lambda to emphasize local details."
However, in the title of Figure 6 on the same page, the description is: "Therefore, we use smaller $\lambda$ values ​​(0.5-0.7) in the shallow layers and larger values ​​(0.8-1.0) in the deeper layers." According to Formula 7, $\lambda$ is the weight of the attention term, so a larger $\lambda$ means more attention to local targets (attention). These two descriptions are completely contradictory. This serious contradiction makes me doubt the implementation details of the paper and the rigor of the author.

2. Heuristic Treatment of "Global Objective": The paper formalizes the "global objective" as the sum of the output errors of all pruned layers (Formula 6), and proposes to use "diversity" as its proxy. This connection is intuitive, but lacks a formal proof. The final optimized objective function (Formula 7) is a weighted sum of attention and diversity scores, rather than a direct optimization of Formula 6. This jump from the true objective to the proxy objective is based on heuristics and lacks stronger theoretical support.

3. Although the ablation experiment in Figure 6 shows the impact of a fixed $\lambda$ value on performance, the paper does not detail how the $\lambda$ schedule across different stages in the final experiment is determined. For example, what is the specific strategy for transitioning from 0.5/0.7 (shallow layers) to 0.8/1.0 (deep layers)? Is this strategy fixed or does it need to be adjusted for different models? This makes the reproducibility of the method and its sensitivity to hyperparameters unclear.

---

> ### Author Rebuttal · Authors · 2025-07-30
>
> We sincerely thank you for your careful review and insightful feedback. We appreciate the opportunity to address the concerns you raised. Below, we provide our responses to the weaknesses:
>
> ---
>
> **W1**: The paper makes completely unclear statements when describing how the core balance factor $ \lambda $ changes with the depth of the network.
> **R1**: Thank you for pointing out the unclear description of the balance factor $ \lambda $ in Section 5.3. We acknowledge a typographical error, which incorrectly states: “in the early layers, we use a larger $\lambda$ value to focus more on global information, while in the deeper layers, we use a smaller lambda to emphasize local details.” In our method, $ \lambda $ increases with network depth. This strategy is correctly reflected in Formula (7), Figure 3, and Section 4.2, and has shown strong empirical performance. We will revise the text for consistency and thank you again for your helpful feedback.
>
> ---
>
> **W2**: Heuristic Treatment of "Global Objective": The final optimized objective function (Formula (7)) is a weighted sum of attention and diversity scores, rather than a direct optimization of Formula (6). This jump from the true objective to the proxy objective is based on heuristics and lacks stronger theoretical support.
> **R2**: Thank you for pointing out this issue which is critical for improving the quality of our paper. We will provide both theoretical and empirical justifications to explain why optimizing Formula (7) effectively leads to the optimization of Formula (6). Suppose token pruning is performed at layers $l_1$ < $l_2$ < $l_3$. Since the tokens retained in deeper layers are a subset of those retained in earlier layers, the preserved token sets satisfy the relationship: $ P_3 \subseteq P_2 \subseteq P_1 $.
>
> **Relation between attention scores and global objective**: The global objective aims to minimize the discrepancy between the hidden states of pruned layers $ l_1, l_2, l_3 $ and those from the original (unpruned) model. According to Formula (1), the hidden state at each pruned layer primarily depends on the outputs of both the attention and MLP modules. Since the MLP parameters are fixed and its input includes the output of the attention module, **we attribute the changes in the pruned layer's hidden states primarily to variations in the attention module's output**. According to Formula (2), the attention module's output is affected by attention score and $V_l$. In Appendix A.1, we observe that different image tokens exhibit similar magnitudes in their value vectors (V). Therefore, maximizing the sum of attention scores contributes to improving the local output similarity, as reflected in the first term of Formula (7). This helps reduce the deviation between the outputs of shadow layers ($l_1$) and the original outputs, thereby contributing to the optimization of the global objective.
>
> **Relation between diversity scores and global objective**: As shown in Figure 1, different layers attend to different regions of the image tokens—that is, the regions with high attention scores vary across layers. Therefore, selecting only high-attention tokens in the shallow layers may lead to local output deviations in subsequent layers. As the diversity objective is strengthened, the overlap between the tokens retained in deeper layers $l_2,l_3$ and the optimal retained token set $ P^* $ increases: $|P^* \cap P^{attn}| < |P^* \cap P^{div}|$. This helps reduce the deviation between the outputs of deeper layers ($l_2,l_3$) and the original outputs, thereby contributing to the optimization of the global objective.
>
> We also observe that the attention from output tokens to image tokens is sparse. To quantify this, we compute the proportion of total attention assigned to image tokens that is captured by the top-k% highest-attention image tokens:
> | Layer | top&nbsp;1% | top&nbsp;5% | top&nbsp;10% | top&nbsp;25% | top&nbsp;50% | top&nbsp;75% | top&nbsp;100% |
> |:------:|:------:|:------:|:------:|:------:|:------:|:------:|:------:|
> |   4   |   0.62   |   0.76   |  0.81    |   0.89   |   0.95   |   0.98   |   1   |
> |   8   |   0.38   |    0.60  |   0.71   |   0.83   |   0.93   |    0.98  |   1   |
> |   12   |   0.32   |   0.56   |   0.69   |    0.84  |   0.93   |   0.98   |   1   |
> |   16   |   0.21   |   0.59   |   0.76   |    0.90  |   0.96   |  0.98    |    1  |
>
> We can see that as the number of retained tokens increases, the gain from optimizing the first term of Formula (7) diminishes, while the benefit of promoting diversity to support subsequent layers increases. In summary, we believe that optimizing Equation (7) serves as a reasonable approximation to optimizing Equation (6). We will include this discussion in Section 4.2 of the revised paper. A more rigorous theoretical justification will be explored in future work.
>
> ---
>
> **W3**: How the schedule across different stages in the final experiment is determined? Is this strategy fixed or does it need to be adjusted for different models? This makes the reproducibility of the method and its sensitivity to hyperparameters unclear.
> **R3**: Thank you for your constructive comment. In our experiments, the $\lambda$ parameter was manually selected. Its value varies across models due to differences in decoding depth and the patterns of image token variation. However, for a given model, the $\lambda$ value remains fixed across different benchmarks. The table below summarizes the $\lambda$ values used for each model at pruning layers:
> | Model | $\lambda$ of different layers |
> |:------:|:------:|
> |   llava-v1.5-7b   |   (0.6,0.8,1.0)  |
> |   llava-v1.5-13b   |   (0.6,0.8,1.0)   |
> |   llava-v1.6-7b  |   (0.4,0.7,1.0)   |
> |   Qwen-2.5vl-7b   |   (0.2,0.5,0.8,1.0)   |
>
> We will include this information in the experimental setup section of Appendix A.2. Once the pruned layers are determined, the $\lambda$ parameter is also fixed. We included preliminary ablation results in Figure 6. To further investigate the sensitivity of our method to the choice of $\lambda$, we conduct an ablation study on LLaVA-v1.5-7B by varying $\lambda$ at different stages. The $\lambda$ values are used in pruning layers $(l_1,l_2,l_3)$. The results are shown below:
> **Experiments in the shallow layer**
> | $\lambda$ | MMB| MME | GQA | POPE| SQA |
> |------|:------:|:------:|:------:|:------:|:------:|
> |   (0.3,0.8,1.0)   |   **63.4**   |   1497   |   **58.9**   |  **85.3**    |   69.1   |
> |   (0.5,0.8,1.0)   |   63.1   |   **1500**   |   58.3   |  84.7    |    **69.6**  |
> |   (0.7,0.8,1.0)   |  _62.7_    |   _1490_   |   _57.9_   |  _84.5_    |   _68.6_   |
>
> **Experiments in the middle layer**
> | $\lambda$ | MMB| MME | GQA | POPE| SQA |
> |------|:------:|:------:|:------:|:------:|:------:|
> |   (0.5,0.6,1.0)   |   **63.2**   |   1486   |   **59.0**   |   **85.2**   |   **69.4**   |
> |   (0.5,0.8,1.0)   |   63.1   |   **1500**   |   58.3   |  84.7    |    69.6  |
> |   (0.5,1.0,1.0)   |   _59.3_   |   _1428_  |  _55.8_    |   _82.9_   |  _54.4_    |
>
> **Experiments in the deep layer**
> | $\lambda$ | MMB| MME | GQA | POPE| SQA |
> |------|:------:|:------:|:------:|:------:|:------:|
> |   (0.3,0.8,1.0)   |   **63.4**   |   **1497**   |   **58.9**   |  **85.3**    |   **69.1**   |
> |   (0.3,0.6,0.8)   |   63.2   |   1491   |   58.5   |  84.3    |    68.9  |
>
> We observe that focusing too early on attention-based pruning in shallow or middle layers leads to performance degradation. Similarly, emphasizing diversity-based pruning in the deeper layers also results in a drop in performance. We will include this discussion in the ablation study section of the revised paper.

---

### Official Review · Reviewer_Gsmq · 2025-07-05

**Clarity:** 3
**Significance:** 3
**Originality:** 2
**Rating:** 4
**Confidence:** 3

**Summary:**

This paper proposes Balanced Token Pruning (BTP), a novel plug-and-play method for accelerating large vision-language models (LVLMs) by reducing visual token redundancy. Unlike prior methods that focus on either local (layer-wise) or global (multi-layer) optimization, BTP introduces a local-global pruning objective that balances preserving immediate output fidelity with maintaining downstream feature diversity. BTP is extensively evaluated on several SOTA LVLMs (e.g., LLaVA-7B/13B, Qwen-VL), achieving significant reductions in token count, latency, and memory, while maintaining ~98% of original performance across multiple benchmarks.

**Questions:**

N/A

**Ethical Concerns:**

["NO or VERY MINOR ethics concerns only"]

**Limitations:**

yes

**Quality:**

3

**Strengths And Weaknesses:**

Strengths.

- The local-global pruning objective is conceptually well-motivated and addresses a clear limitation in previous token pruning strategies. The balance factor $\lambda$ and stage-wise optimization strategy are intuitive and theoretically grounded.
- Across multiple models and datasets (GQA, MME, POPE, SQA, MM-VeT), BTP consistently outperforms state-of-the-art approaches like FastV, DivPrune, and VTW in both performance retention and efficiency.
- The authors show that BTP is effective across different model architectures and scales (LLaVA-7B/13B, Qwen-VL), increasing its impact and generalization potential.

Weaknesses.

- The method relies on a calibration set to determine pruning layers and balance factors. While the authors state that the same calibration set is reused across experiments, further discussion is needed on how sensitive the method is to the size and content of this set.
- It is not clear how Figure 2 is depicted, e.g., which data and features are used to measure similarity? It would be good to provide more details.

---

> ### Author Rebuttal · Authors · 2025-07-30
>
> Thank you for your insightful feedback. We appreciate the opportunity to address the concerns you raised. Below, we provide our responses to the weaknesses:
>
> ---
>
> **W1**: The method relies on a calibration set to determine pruning layers and balance factors. How sensitive the method is to the size and content of this set?
> **R1**: Thank you for your constructive comment. In our approach, the selection of pruned layers is guided by the calibration set. The trade-off parameter $ \lambda $ is a fixed hyperparameter and is not determined by the calibration set. Our method is insensitive to the content and size of the calibration set.
>
> **Insensitive to the text content**: According to Equation (2) in the paper, in multimodal large language models, *M* denotes the causal mask, which constrains each token to attend only to preceding tokens. In our input format, image tokens always precede the text prompts (i.e., the input follows the structure: prefix + image + question). As a result, the model processes the image tokens before it receives the specific question which is unrelated to the input question. To validate our hypothesis, we posed different types of questions on the same image. We then conducted the experiment presented in Figure 4 using LLaVA-v1.5.  We computed the number of image tokens whose cosine similarity between adjacent layers falls below 0.93. The resulting trends are shown as follows:
> | Question_type | layer1 | layer3 | layer5 | layer7 | layer9 | layer11 | layer13 | layer15 | layer17 | layer19 | layer21 | layer23 | layer25 | layer27 | layer29 |
> |------|:------:|:------:|:------:|:------:|:------:|:------:|:------:|:------:|:-------:|:-------:|:-------:|:-------:|:-------:|:-------:|:-------:|
> |  Caption    |   0   |    0  |   0   |   151   |  325    |   359   |   141   |   136   |   155    |  38     |   45    |    11   |    1   |   0    |    3   |
> |   POPE   |   0   |    0  |   0   |   151   |  325    |   359   |   141   |   136   |   155    |  38     |   45    |    11   |    1   |   0    |    3   |
> |   SpatialVQA   | 0   |    0  |   0   |   151   |  325    |   359   |   141   |   136   |   155    |  38     |   45    |    11   |    1   |   0    |    3   |
>
> We observe that varying the question type for the same image does not lead to significant differences in the results. In the following analysis, we investigate how image content and the size of the calibration set affect our method.
>
> **Insensitive to the image content and size**: We expanded the calibration set by incorporating images from multiple datasets, including GQA, V*Bench[1], and SQA. We then repeated the experiment shown in Figure 4 using calibration set sizes of 64, 128, and 256. Specifically, we computed the number of image tokens whose cosine similarity between adjacent layers falls below 0.93 using LLaVA-v1.5. The results are presented below:
> | Size | layer1 | layer3 | layer5 | layer7 | layer9 | layer11 | layer13 | layer15 | layer17 | layer19 | layer21 | layer23 | layer25 | layer27 | layer29 |
> |------|:------:|:------:|:------:|:------:|:------:|:------:|:------:|:------:|:-------:|:-------:|:-------:|:-------:|:-------:|:-------:|:-------:|
> |  64    |   0   |    0  |   0   |   151   |  325    |   359   |   141   |   136   |   155    |  38     |   45    |    11   |    1   |   0    |    3   |
> |   128   |   0   |    0  |   0   |   164   |  350   |   385  |   166  |   139  |   127    |  65     |   36    |    16   |    5   |   0    |    2   |
> |   256   | 0   |    0  |   0   |   132  |  332  |   368 |   174  |   155   |   145    |    61   |   32    |    11  |    3   |   0    |    1  |
>
> We observe that the patterns remain consistent across different image contents and calibration set sizes, indicating that these factors do not affect the behavior of our method. We will include this additional analysis in the Ablation Study section (Section 5.3) of the revised paper.
>
> ---
>
> **W2**: It is not clear how Figure 2 is depicted, e.g., which data and features are used to measure similarity? It would be good to provide more details.
> **R2**: Thank you for your constructive feedback. We conducted experiments on the captioning task. In Figure 2, our goal is to investigate the impact of pruning during the prefilling stage on the model’s output. In multimodal large language models, the first token generated during the decoding stage is directly influenced by the final token from the prefilling stage. This final token is also the only point during prefilling where the model produces an output. Therefore, we evaluate the effect of different pruning strategies by analyzing their impact on the hidden state of the last token in the prefilling stage.
>
> **Effect of Different Pruning Strategies on the Full Output**: In the captioning task, multimodal large language models generate not just a single token, but a sequence of tokens in an autoregressive manner. Therefore, we further investigated how different pruning strategies affect tokens at various positions in the output sequence. Specifically, we analyzed the impact on the 1st, 4th, and 20th tokens. We use llava-v1.5-7b to conduct this experiment. The results are summarized below:
> | Method-Position | Layer4 | Layer8 | Layer12 | Layer16 | Layer20 | Layer24 |
> |------|:------:|:------:|:------:|:------:|:------:|:------:|
> |   Attn(1st)   |   1   |    1  |  **0.996**    |   **0.994**   |  0.979    |   0.973   |
> |   Div(1st)   |    1  |    1  |  0.992    |   0.991   |    **0.985**  |   **0.982**   |
> |   Attn(4th)   |   1   |   **0.998**  |   **0.993**   |   **0.991**   |   0.954   |  0.949    |
> |   Div(4th)   |   1   |   0.995 |    0.989  |    0.989  |   **0.975**   |  **0.972**    |
> |   Attn(15th)   |   **0.369**   |  **0.311**  |  **0.375**   |   0.279   |  0.303 |   0.352   |
> |   Div(15th)  |   0.299   |   0.204   |  0.274   |   **0.312**  |  **0.423**   |    **0.496**  |
> |   Attn(20th)   |   **0.297**   |  0.213  |  0.141   |   0.161   |  0.168 |   0.166   |
> |   Div(20th)  |   0.292   |   **0.255**   |  **0.162**    |   **0.171**   |  **0.169**    |    **0.235**  |
>
> We observe that tokens appearing later in the output sequence are influenced by previously generated tokens, leading to a significant drop in similarity with the corresponding tokens from the original (non-pruned) output. Moreover, **tokens at different positions exhibit similar patterns under both pruning strategies**: attention based methods focus on local goal and diversity based methods focus on global goal.These findings further support the validity of our observations. We will include this discussion in the Introduction section of the revised manuscript.
>
> [1] V*: Guided Visual Search as a Core Mechanism in Multimodal LLMs.

---

> > ### Comment · Reviewer_Gsmq · 2025-08-06
> >
> > Thank the authors for their detailed responses. The rebuttal provides additional experiments to validate the method, which address my concerns. I would like to maintain my positive rating.

---

### Note · Authors · 2025-08-12

Dear Reviewers and Area Chairs,

Thank you for your diligent work and insightful feedback throughout the rebuttal and discussion process. We are pleased to have the opportunity to provide a brief summary of our rebuttal and the discussions.

We are pleased that all reviewers **gave encouraging evaluations and  positive scores (4,4,4,4) in the first round**:
1. The paper formulated a **novel pruning objective function** and proposing an **intuitive and very novel pruning strategy**. (Reviewer #Gsmq, Reviewer #W7Hn, Reviewer #wCTc)
2. The paper provided **strong empirical analyses**, which which strongly supports our analysis and method design. (All Reviewers)
3. The paper made extensive experiments on multiple state-of-the-art models and datasets, which enhances the **impact and generalization potential of our method**.  (All Reviewers)

We have added more details on the empirical and performance experiments, conducted more extensive ablation studies, and provided further theoretical analysis in the discussion period. We are pleased that the **all reviewers stated their concerns have been well solved during the rebuttal and discussion stage**:
1. Reviewer Gsmq stated that our additional explanations regarding the calibration set analysis and exploratory experiments have addressed their concerns: ***"I would like to maintain my positive rating."***
2. Reviewer wCTc gave positive feedback towards our further analysis of the experimental phenomena and the explanations of our method design: ***"Overall, the clarifications and additional results make your work stronger. I have no further concerns."***
3. Reviewer jfqF appreciated the additional ablation studies we conducted: ***"Your response has addressed most of my concerns, and I now have a much better understanding of your design choices and results."***
4. Reviewer W7Hn confirmed our responses via submitting Mandatory Acknowledgement. During the rebuttal phase, we have tried our best to address his/her concerns on hyperparameters and the objective function approximation with new experimental results and theoretical derivations (these questions are very similar to Reviewer jfqF and Reviewer jfqF gave positive feedback on our response). Thus, although Reviewer W7Hn did not provide additional comments, we believe the acknowledgment reflects the approval.

Thank you again for your efforts and contributions toward a fair and insightful discussion.

---

### Decision · Program_Chairs · 2025-09-17

**Decision:**

Accept (poster)

**Comment:**

All four reviewers initially rated the submission as borderline accept (4), and the discussion and rebuttal phase led to one reviewer raising their score to 5. The paper presents a novel and well-motivated method—Balanced Token Pruning (BTP)—to reduce computational costs in LVLMs by balancing local and global pruning objectives across layers. Reviewers found the approach intuitive, technically sound, and empirically well-supported, especially appreciating the extensive ablation studies and thorough rebuttal. While minor concerns remain (e.g., hyperparameter tuning clarity and practical batching implications), the authors addressed these with detailed analysis. Given the overall positive assessments and the clarity brought through rebuttal and discussion, I recommend acceptance.